# Epi-Brassinolide Regulates *ZmC4 NADP-ME* Expression through the Transcription Factors *ZmbHLH157* and *ZmNF-YC2*

**DOI:** 10.3390/ijms24054614

**Published:** 2023-02-27

**Authors:** Yuanfen Gao, Xuewu He, Huayang Lv, Hanmei Liu, Yangping Li, Yufeng Hu, Yinghong Liu, Yubi Huang, Junjie Zhang

**Affiliations:** 1College of Life Science, Sichuan Agricultural University, Ya’an 625000, China; 2State Key Laboratory of Crop Gene Exploration and Utilization in Southwest China, Sichuan Agricultural University, Chengdu 611130, China; 3Maize Research Institute, Sichuan Agricultural University, Chengdu 611130, China

**Keywords:** bHLH157, *C4 NADP-ME*, epi-brassinolide, NF-YC2, photosynthesis

## Abstract

Maize is a main food and feed crop with great production potential and high economic benefits. Improving its photosynthesis efficiency is crucial for increasing yield. Maize photosynthesis occurs mainly through the C4 pathway, and NADP-ME (NADP-malic enzyme) is a key enzyme in the photosynthetic carbon assimilation pathway of C4 plants. ZmC4-NADP-ME catalyzes the release of CO_2_ from oxaloacetate into the Calvin cycle in the maize bundle sheath. Brassinosteroid (BL) can improve photosynthesis; however, its molecular mechanism of action remains unclear. In this study, transcriptome sequencing of maize seedlings treated with epi-brassinolide (EBL) showed that differentially expressed genes (DEGs) were significantly enriched in photosynthetic antenna proteins, porphyrin and chlorophyll metabolism, and photosynthesis pathways. The DEGs of *C4-NADP-ME* and pyruvate phosphate dikinase in the C4 pathway were significantly enriched in EBL treatment. Co-expression analysis showed that the transcription level of ZmNF-YC2 and ZmbHLH157 transcription factors was increased under EBL treatment and moderately positively correlated with *ZmC4-NADP-ME*. Transient overexpression of protoplasts revealed that ZmNF-YC2 and ZmbHLH157 activate *C4-NADP-ME* promoters. Further experiments showed ZmNF-YC2 and ZmbHLH157 transcription factor binding sites on the −1616 bp and −1118 bp ZmC4 NADP-ME promoter. ZmNF-YC2 and ZmbHLH157 were screened as candidate transcription factors mediating brassinosteroid hormone regulation of the ZmC4 NADP-ME gene. The results provide a theoretical basis for improving maize yield using BR hormones.

## 1. Introduction

As one of the important crops in the world, the yield of maize is only lower than wheat and rice. Enhancing photosynthetic capacity is one of the ways to increase maize yield [1]. Photosynthesis is a crucial process in green plants; therefore, improving the efficiency of photosynthesis is one of the ways of increasing yield [2]. Maize photosynthesis occurs mainly through the C4 pathway. Unlike C3 plants, C4 plants first form malic acid by phosphoenolpyruvate (PEP) and CO_2_ in mesophyll cells [3]. Malic acid is transferred to sheath cells and releases CO_2_ by decarboxylase, which enters the photosynthetic carbon cycle via ribulose diphosphate (RuBP) carboxylase in the chloroplasts of sheath cells [4]. Therefore, decarboxylase is crucial for CO_2_ fixation in maize, and it is of great significance to study the regulation of the expression of its encoding genes to improve photosynthetic efficiency. 

Three types of decarboxylases catalyze deacidification of malic acid, namely NADP-dependent malic enzyme (NADP-ME), NAD-dependent malic enzyme (NAD-ME), and phosphoenolpyruvate carboxykinase (PCK) [5], and the decarboxylase in maize is mainly NADP-ME [6]. Three NADP-ME isoforms have been identified in maize: one is the plastid type (C4 NADP-ME) involved in C4 photosynthesis, and it is subcellularly localized in the chloroplast [7]; the other is the plastid non-phototype (C4 non-NADP-ME) that responds to plant defense inducers [8]; and the third is the cytoplasmic subtype (CYT-NADP-ME), which is enriched in embryos and roots and participates in the control of cytosolic acid levels [7]. Although the three NADP-MEs are similar, the expression level of ZmC4 NADP-ME in the leaves is 20 times more than that of cytoplasmic NADP-ME [9]. Overexpression of maize *C4-NADP-ME* in tobacco increases the efficiency of net CO_2_ fixation, and this transgenic tobacco exhibited a higher phenotype [10], and overexpression of *SbC4-NADP-ME* in sweet sorghum, which is a C4 plant, has a positive effect on its photosynthetic capacity.

Brassinosteriods (BRs), a sterol hormone type, have the ability to affect plant photosynthesis. More than 70 BRs have been identified to date, with 24-epibrassinolide (24-EBL) and 28-homobrassinolide (28-homobrassinolide, 28-HBL) being the most active and commercialized [11]. BRs can promote CO_2_ fixation and dark reactions in plant photosynthesis [12], and studies have demonstrated that exogenous spraying or overexpression of endogenous brassinolide (BL) on crops, such as soybean [13], *Brassica juncea* [14], *Triticum aestivum* [15], rice [16], cucumber [17], and tomato [18], has improved the transcription level and activities of dark reaction enzymes, and promoted CO_2_ assimilation, resulting in an increase in photosynthetic products. In fact, EBL regulates photosynthesis through different pathways. EBL affects stomatal development by regulating the bHLH family transcription factor SPEECHLESS, and further has an impact on plant photosynthesis [19]. The BES1(BRINSENSITIVE 1-EMS-SUPPRESSOR1) mediated the regulation of EBL on chloroplast development [20]. However, the regulation of transcription factors mediated by EBL on carbon fixation is not clear.

It has been reported that the transcription levels of *RCA* (*rubisco activase*) and *fructose-1,6-bisphosphatase* (*FBPase*) are promoted by BR treatment [21,22]. However, whether BL has an effect on the expression of the *C4-NADP-ME* gene in C4 plants and the possible mechanisms is still poorly understood. In this experiment, the transcriptome analysis of maize leaves treated with EBL was performed, and it was confirmed that EBL played a role in the differential expression of maize leaf photosynthesis genes. We studied the effect of EBL on the expression of the *ZmC4-NADP-ME* gene in maize, and screened and identified two transcription factors that might mediate BL regulation of the expression of *ZmC4-NADP-ME*. 

## 2. Results

### 2.1. Transcriptome Data Analysis of EBL-Treated Maize Leaves

To determine the genes regulated by EBL in maize seedlings, DEGs were analyzed following EBL treatment. The results showed that 2041 DEGs were observed in 0.5 μM EBL treatment, of which 1051 DEGs were upregulated and 990 DEGs were downregulated (Figure 1, Appendix A). The KEGG pathway analysis was performed in DEGs to identify the enriched pathways. The five main KEGG pathways were photosynthetic antenna protein, ribosome, porphyrin and chlorophyll metabolism, photosynthesis, and synthesis of secondary metabolites (Figure 1), indicating that EBL treatment affects plant chlorophyll and photosynthetic electron transport [20]. The carbon fixation pathway was not the main enriched pathway in KEGG, but the results also showed that DEGs in the EBL treatment were significantly enriched in the carbon fixation pathway (*p* < 0.05) (Appendix A).

### 2.2. The Impact of EBL on C4-NADP-ME Expression in Maize Leaves

Previous experiments have shown that exogenous spraying of EBL promotes the transcription of dark reaction enzymes [23]. We found that the expression of *ZmC4 NADP-ME* following EBL treatment was upregulated in RNA-seq (Figure 2A). To further study how EBL regulates *ZmC4 NADP-ME* expression, different concentrations of EBL were used to spray maize seedlings or whole plants in the field. The results showed that the expression of the *ZmC4 NADP-ME* gene significantly increased by 1.8 times on the 5th day after 150 nmol/L EBL in the field. On the 10th day after treatment with 100 nmol/L EBL, the expression of the gene increased by approximately seven times. The expression of *ZmC4 NADP-ME* in the leaves of maize seedlings increased significantly by 8.6 times after 24 h EBL treatment (Figure 2B). These results show that EBL can promote the expression of *C4-NADP-ME* in maize leaves.

### 2.3. bHLH157 and NF-YC2 Are Induced by EBL and Affect ZmC4-NADP-ME Expression

We screened for transcription factors involved in the regulation of *ZmC4 NADP-ME* expression mediated by EBL. The upregulated transcription factors in the transcriptome data were selected, and Pearson correlation coefficients (PCCs, R) were calculated with the reference transcriptome in leaves at different stages provided by Stelpflug [24] (Appendix A). Fluorescence quantification was performed to verify the transcription levels of transcription factors after EBL treatment. The bHLH and NF-Y family transcription factors that showed enhanced expression after EBL treatment and may be involved in carbon fixation were selected (Figure 2C) [25]. The R coefficients between ZmC4 NADP-ME and bHLH157 or NF-YC2 were 0.66 and 0.75, respectively (Appendix A). This indicated that *ZmC4 NADP-ME* was moderately positively (0.5 ≤ |r| < 0.8) correlated with the two transcription factors, and the expression levels of *bHLH157* and *NF-YC2* were significantly increased by 2.7 and 9.3 times, respectively, after exogenous EBL treatment (Figure 2C). Moreover, the dual-luciferase system showed that bHLH157 and NF-YC2 promoted the expression of genes downstream of the C4-NADP-ME promoter (Figure 3A). The yeast one-hybrid assay showed that bHLH157 and NF-YC2 in yeast combined with the *C4-NADP-ME* promoter to make the experimental group of yeast grow better on the SD/-Leu/-Trp/-His medium with 150 mmol/L 3-AT than the control (Figure 3C,D). We also verified the synergy between bHLH157 and NF-YC2; when bHLH157 and NF-YC2 were transformed together, the promoter activity was further improved compared to when they were transformed separately (Figure 3B). However, yeast two-hybrid experiments showed that bHLH157 and NF-YC2 did not interact in yeast cells (Figure 3E).

### 2.4. Functional Verification of bHLH157 and NF-YC2 Transcription Factor

The subcellular localization and self-activation activity were used to determine the transcription factor characteristics of bHLH157 and NF-YC2. NF-YC2 and bHLH157 were located in the nuclei of maize leaf cells (Figure 4A). The results of yeast self-activation indicated that ZmbHLH157 and ZmNF-YC2 have transcriptional activation activities (Figure 4B). Additionally, the expression patterns of *ZmC4 NADP-ME*, *ZmbHLH157* and *ZmNF-YC2* in different parts of maize were analyzed to explore the temporal and spatial specificity of gene expression. The results showed that *ZmbHLH157* and *ZmNF-YC2* have higher expression levels in leaves (Figure 4C).

### 2.5. Preliminary Study on the Binding Region of bHLH157, NF-YC2 and ZmC4-NADP-ME Promoter

To further study the binding sites between the transcription factors, bHLH157 and NF-YC2 and the promoter of ZmC4 NADP-ME, Plant Care (https://bioinformatics.psb.ugent.be/webtools/plantcare/html/, accessed on 22 March 2019) was used to analyze the cis-acting elements in the promoter region [26]. To preliminarily explore the possible binding regions, a dual luciferase system was used to verify the transactivation of bHLH157 and NF-YC2 on ZmC4 NADP-ME promoters of different lengths. 

The cis-acting element analysis results indicated that the binding site of cis-acting element E-box that bHLH157 may bind to is distributed in the ZmC4 NADP-ME promoter at −1152 bp, −669 bp and −435 bp (Table 1). The cis-acting element CCAAT-box that NF-YC2 may bind to was mainly distributed at the −1495 bp promoter (Table 1). When the length of the ZmC4 NADP-ME promoter was 1118 bp, the effect of bHLH157 and NF-YC2 on the relative activity of LUC was significantly reduced in comparison to a length of 1616 bp, indicating that the region from −1616 bp to −1118 bp was the binding region of bHLH157 and NF-YC2 (Figure 5A–C). We speculated that NF-YC2 may bind to the CCAAT-box or CAAT-box in this region, while bHLH157 may bind to the G-box in this region, and the remaining sites are presumably related to the constitutive activity of the promoter.

### 2.6. EBL Further Enhances ZmC4 NADP-ME Promoter Activity by increasing bHLH157 and NF-YC2 Expression

To determine whether the effect of EBL on the expression of bHLH157 and NF-YC2 further activated the *ZmC4-NADP-ME* promoter, EBL was added to the protoplasts. The results showed that the transactivation effect of bHLH157 and NF-YC2 on the *ZmC4-NADP-ME* promoter was significantly increased by 1.55 and 0.79 times, respectively, compared with the control after adding 1 nmol/L EBL for 12 h (Figure 5D). In contrast, EBL treatment had no effect on the expression of the LUC reporter under the Ubi promoter, indicating that the promotion effect of EBL on the expression of *ZmC4-NADP-ME* in maize protoplasts was related to the transcription factor bHLH157 or NF-YC2.

## 3. Discussion

BR has previously been found to indirectly affect plant photosynthesis by controlling stomatal size and chloroplast development [11,27]. We found that the DEGs of EBL treatment were enriched in the photosynthetic antenna protein, and porphyrin and chlorophyll metabolism pathway of maize leaves. BR affects chlorophyll content in *Brassica juncea* [11]. And we have previously found that chlorophyll content in maize leaves is affected by EBL treatment [23]. The light-harvesting complex in chloroplast mainly consists of photosynthetic antenna protein and chlorophyll. It has been found that BRs play a role in the regulation of chloroplast development [20]. We speculate that EBL differentially regulates genes in the photosynthetic antenna protein and chlorophyll metabolism pathway in leaves, which is an important factor affecting photosynthesis in chloroplast. Many studies have shown that bHLH family transcription factors can participate in the expression of carbon fixation-related enzyme genes to regulate photosynthesis [28]. Transcription factors of the bHLH family, such as BES1 (BRASSINOSTEROIDROIDISTTIVE 1-EMS-SUPPRESSOR 1) and BZR1 (BRASSINAZOLE RESISTANT 1), are important transcription factors in BRs signal transduction, and E-box (CANNTG), a cis-acting element response to BRs, is a putative binding element of the bHLH family [29]. This experiment showed that the bHLH157 transcription factor activates the promoter of ZmC4 NADP-ME. The transcription factor was highly expressed in maize leaves and responded to exogenous spraying. Moreover, bHLH157 transcription factors may combine with the G-box in the C4-NADP-ME promoter region, which is a common binding region of the bHLH family transcription factors [30]. Moreover, it has been found that certain transcription factors in the bHLH family can bind to the G-box of the C4-NADP-ME promoter [28]. The bHLH157 transcription factor transferred into protoplasts were treated with EBL, which had no affect on the relative LUC activity; however, when the bHLH157 and C4-NADP-ME promoter were co-transferred and treated with EBL, the relative value of LUC increased by 1.55 times. Presumably, a low concentration of EBL further promoted the Ubi::bHLH157 and C4-NADP-ME promoter interaction, and this may be related to the regulation of protein synthesis or modification by EBL.

The F-Y family transcription factors are also involved in the regulation of dark reaction enzymes [31]. The three different subunits in the NF-Y family (NF-YA, NF-YB, and NF-YC) generally form a complex that binds to the CCAAT cis-acting element in the promoter and can also interact with transcription factors of other families to regulate gene expression [32], but only a few active NF-Y complexes are known in plants [33]. In this study, ZmNF-YC2 was found to be a nuclear localization protein. The self-activation verification of NF-YC2 showed that the NF-YC2 subunit alone was self-activated, but only the overexpression of the NF-YC2 subunit in protoplasts still increased the activity of the *ZmC4-NADP-ME* promoter. We attribute this phenomenon to the correlation between gene expression levels. A correlation between the expressions of related proteins was also found in rice. In transgenic rice plants with antisense or RNAi of the chloroplast biogenesis regulatory gene *OsHAP3A*, the expression of *OsHAP3A* was reduced, and *OsHAP3B* and *OsHAP3C* formed a complex with it [31]. Transcription factors of the NF-Y family have been found to be involved in BL signal transduction, such as LEAFY COTYLEDON1 (LEC1) [33]. Overexpression of *LEC1* would inhibit the negative regulator BRH1 of BR signaling, and the gene *DOGT1* related to BR catabolism [34]. Notably, Su et al. discovered *ZmNF-YC2* is a positive regulator of maize flowering time under long-day conditions [35]. We speculate that NF-YC2 may be involved in multiple light-related physiological processes with powerful functions. 

Previous studies have shown that NF-Y family proteins can interact with other families; for example, OsbZIP76 in rice interacts with OsNF-YBs to regulate endosperm development [36]; and the NF-YB1-YC12-bHLH144 ternary complex regulates the key granule-bound starch synthase gene Wx (granule-bound starch synthase) [33]. In this study, we found that bHLH157 and NF-YC2 synergistically promoted the activity of the *ZmC4-NADP-ME* promoter compared with that of each single promoter (Figure 3B). However, bHLH157 and NF-YC2 did not interact in yeast (Figure 3E). We speculate that there is a third transcription factor involved in this binding [37]. Future studies should explore the function of the bHLH157 and NF-YC2 transcription factors in EBL signaling, which could help to understand in detail the mechanism of EBL in enhanced plant photosynthetic capacity.

## 4. Materials and Methods

### 4.1. Maize Materials, Growth Conditions and EBL Treatment

In this experiment, the maize inbred line Mo17 was used as the experimental material, and the seeds were provided by Sichuan Agricultural University, Sichuan China. Mo17 seeds were sown in peat soil (peat soil: vermiculite = 1:1) and the day and night culture conditions were: 25 °C, 3000 lx light for 16 h, and 23 °C, dark, for 8 h. Maize in the field was planted in Ya’an City, Sichuan Province. In the field, maize plants at the 8-leaf stage were sprayed with EBL all over the plant at concentrations of 50, 100, 150, and 500 nmol/L once every five days, with each group comprising five plants. The sixth leaf of maize in the field at the 8-leaf stage was selected for quantitative real-time PCR (qRT-PCR) with two biological replicates [38]. Six-leaf stage roots (roots, R), stems (St), leaves (leaves, L), filaments (F), anthers (A), pollen (pollen, P), post-pollination seeds (S), embryos (Em), and endosperm (En) in maize were used for semi-quantitative analysis of the relative expression of transcription factors [39]. Cellulase and pectinase were used in the enzymatic cell wall to produce protoplasts. These cells were then treated with 0.02 nmol/L and 0.1 nmol/L EBL, including three biological replicates [40]. Primers were designed using Primer Premier 5.0; the primer sequences are presented in the Appendix A. 

### 4.2. High-Throughput RNA-seq

The isolated maize leaves (the two leaves of a one-leaf seedling are lightly divided by a sharp blade into segments of about 2 cm) used for RNA-seq were treated with 1 mL of 0.5 μmol/L EBL on 1/2MS medium (Solarbio, Beijing, China) for 6 h, and distilled water was used as the control. Leaves treated with 0, 0.2, and 0.5 μmol/L EBL, respectively, were cultured for 24 h. The content of EBL was chosen by a previous study [23].

According to the manufacturer’s protocol, the Macherey-Nagel NucleoSpin miRNA isolation kit (Solarbio, Beijing, China) was used to extract RNA from different treated maize leaves, and these RNA (>200 nt) components were used for RNA-seq (novogene, Beijing, China). The original data were filtered using HISAT software and compared with the reference genome (ftp://ftp.ensemblgenomes.org/pub/plants/release-41/fasta/zea_mays/dna/, accessed on 28 December 2018). The expected number of fragments per kilobase of transcript sequence per million base pairs sequenced (FPKM) was evaluated, and the gene expression levels of each sample were analyzed using the union model in the HTSeq software [38]. The Kyoto Encyclopedia of Genes and Genomes (KEGG) pathway of differentially expressed genes (DEGs) was significantly enriched [41].

### 4.3. Transcription Activation and Protein Structure Prediction

The ClonExpress^®^II One-Step Cloning Kit (Vazyme, Nanjing, China) was used to recombine bHLH157 or NF-YC2 with the pGBKT7 vector. The recombinant plasmid was transformed into AH109 yeast cells and cultured at an OD_600_ of 0.5. Transformed yeasts were diluted 10^−1^/10^−2^/10^−3^/10^−4^ times, then spotted on the surface of an SD/-Trp or SD/-Trp/-His/X-α-gal solid medium, and cultured at 28 °C for 3 days [39].

### 4.4. Subcellular Localization

The pCAMBIA2300-35S-EGFP vector was chosen to carry the bHLH157or NF-YC2 genes with the stop codon removed. The construct concentration for protoplast transformation was measured using the NanoDrop2000 (Thermo Scientific, Waltham, MA, USA). The construct was transformed into protoplasts of maize leaf cells by PEG-mediated transformation and cultured for 12 h at 25 °C under 2000 lx light intensity [42]. A laser confocal microscope (FV1200, OLYMPUS, Shinjuku City, Japan) was used to observe the location of bHLH157 or NF-YC2.

### 4.5. Yeast One-Hybrid

The pGADT7-Rec2 vector carrying the CDS sequence of bHLH157 or NF-YC2 and the pHIS2 vector carrying the promoter of C4 NADP-ME were co-transferred to the Y187 yeast competent and then expanded to an optical density at 600 nm (OD_600_) of 0.5. The yeast concentration was diluted with sterile water by 1, 10^−1^, 10^−2^, 10^−3^, 10^−4^ times. Then, 5 μL of the sample was added to SD/-Leu/-Trp/-His solid medium containing 0, 50, 100, and 150 mmol/L 3-AT.

### 4.6. Protoplasts Transformation and Dual Luciferase System

PEG-Ga solution (40% PEG 4000, 0.2 M Mannitol, 0.1 M GaCl_2_) was used by protoplasts transformation. The single sample needed 110 μL PEG-Ga solution, and this transformation reaction lasted six minutes before dilution with 440 μL MMG (15 mM MgCl_2,_ 0.4 M Mannitol, 4 mM MES). Protoplasts after transformation reaction were washed by WI (0.5 M Mannitol, 4 mM MES, 20 mM KCl) for dual-Luciferase assay. The C4-NADP-ME (Zm00001eb121470) promoter sequence was recombined with the PBI221 vector, in which the GUS (β-glucuronidase) reporter gene was replaced with the LUC (fireflyluciferase) reporter gene [43]. The coding sequences of bHLH157 and NF-YC2 were cloned into the PBI221 vector, and the two recombinant vectors were co-transformed into maize leaf protoplasts to evaluate the relative activity of LUC. Dual-luciferase activity (Promega) was measured to verify the interactions between bHLH157 and/or NF-YC2 and *C4 NADP-ME*. GUS activity was used as an internal reference to normalize the transformation efficiency of protoplasts, and the relative ratio of LUC/GUS (4 h–0 h) between the experimental group and control was used to represent the relative activity of the promoter [44]. 

### 4.7. Data Analysis

SPSS 22.0 was used to process the data, and Tukey’s test was performed to analyze significant differences among groups.

## 5. Conclusions

In this study, transcriptome of maize seedlings treated with EBL indicated the role of EBL in metabolic pathways related to maize leaf photosynthesis. Genes in photosynthetic antenna proteins, porphyrin and chlorophyll metabolism are regulated by EBL. We found *ZmC4-NADP-ME* was upregulated by EBL, and the ZmNF-YC2 and ZmbHLH157 transcription factors were moderately positively correlated with *ZmC4-NADP-ME* by co-expression analysis. Transcription factors ZmNF-YC2 and ZmbHLH157 can promote ZmC4 NADP-ME promoter activity, and have a certain degree of superposition effects. Further research showed that both ZmNF-YC2 and ZmbHLH157 transcription factor binding sites on the promoter of ZmC4 NADP-ME were located between −1616 bp and −1118 bp of the target promoter. EBL is involved in regulating multiple metabolic pathways related to photosynthesis. The results provide a theoretical basis for improving maize yield using BL hormones.

## Figures and Tables

**Figure 1 ijms-24-04614-f001:**
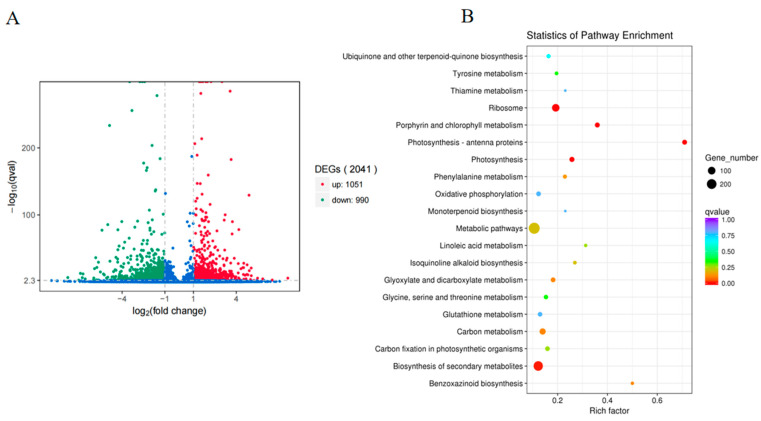
DEGs number and KEGG pathway enrichment with EBL treatment in RNA-seq. (**A**) Volcano plot with DEGs number. (**B**) KEGG pathway scatterplot, rich factor, the ratio of the number of differential genes in the metabolic pathway to the number of all genes annotated to the pathway.

**Figure 2 ijms-24-04614-f002:**
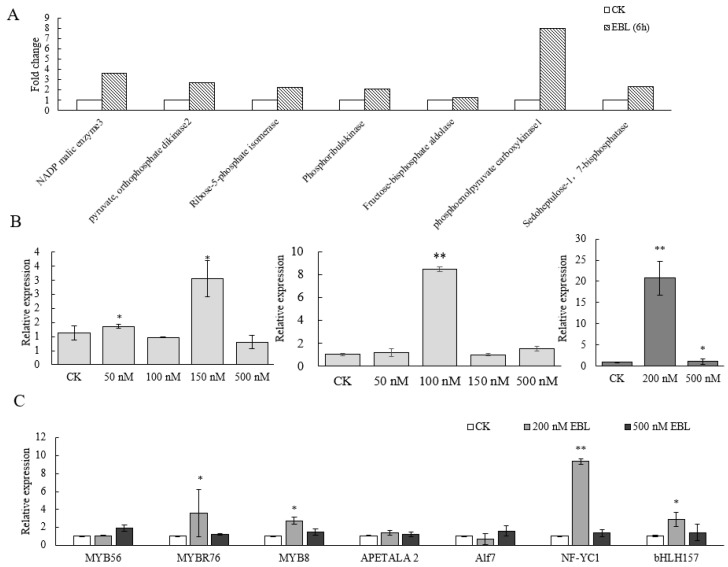
Expression levels of photosynthesis-related genes under EBL treatment. (**A**) The fold change of upregulated carbon fixation enzymes in transcriptome with EBL treatment. NADP malic enzyme 3 (Zm00001d000316), pyruvate phosphate dikinase (Zm00001d010321), ribose 5-phosphate isomerase (Zm00001d013135), phosphoribulokinase (Zm00001d017711), phosphoenolpyruvate carboxykinase (Zm00001d028471), sedoheptulose-1,7-bisphosphatase (Zm00001d042840). (**B**) The transcription level of *ZmC4-NADP-ME* with EBL treatment in the field (maize in 8-leaves stage) on the 5th day and 10th days; the expression of *ZmC4 NADP-ME* in the leaves (seedlings in 2 leaves and 1 core stage). (**C**) Transcription level of the preliminary screened transcription factors treated with EBL (nM). “*” represented as *p* < 0.05, “**” represented as *p* < 0.01.

**Figure 3 ijms-24-04614-f003:**
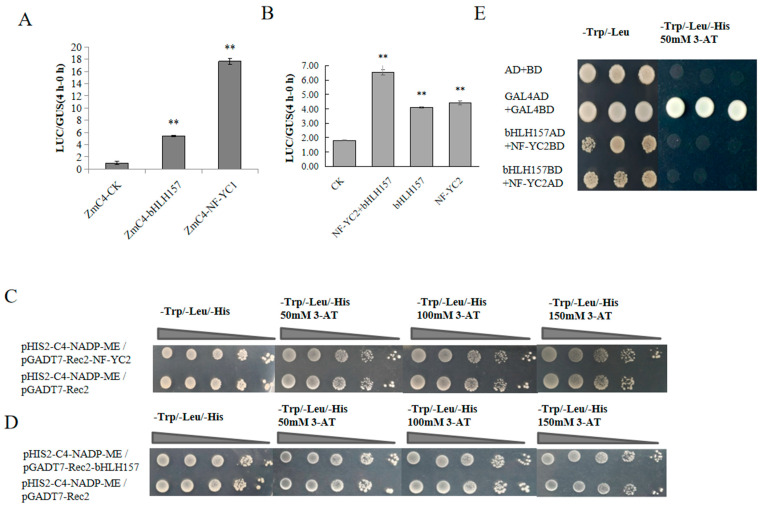
Interaction analysis of bHLH157, NF-YC2 and ZmC4 NADP-ME promoter. (**A**) Interaction with promoter individually in protoplasts. (**B**) Synergistical interaction with promoter. (**C**) The interaction analysis of *NF-YC2* and the promoter in yeast, pGADT7-Rec2, was the negative control. (**D**) The interaction analysis of *bHLH157* and promoter in yeast. (**E**) Two-hybrid in yeast, “AD + BD“ was the negative control and “GAL4AD + GALBD” was the positive control. “**” represented as *p* < 0.01.

**Figure 4 ijms-24-04614-f004:**
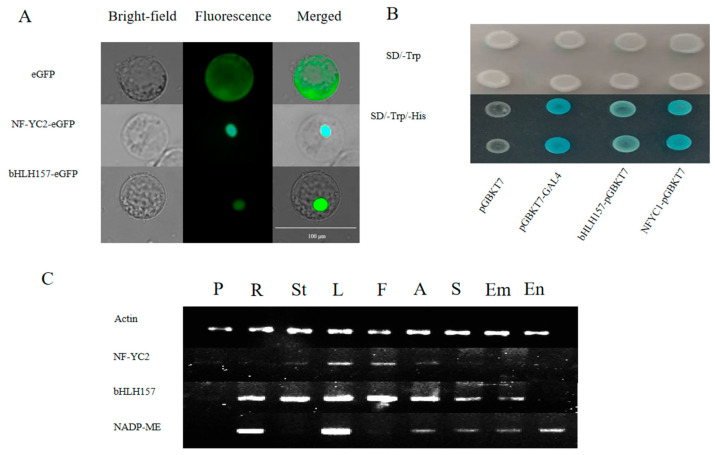
bHLH157 and NF-YC2 transcription factor functional verification. (**A**) Subcellular localization. (**B**) Yeast self-activation, pGBKT7, was the negative control and pGBKT7-GAL4 was the positive control. (**C**) Semi-quantitative, (roots, R), stems (St), leaves (leaves, L), filaments (F), anthers (A), pollen (pollen, P), post-pollination seeds (S), embryos (Em), and endosperm (En).

**Figure 5 ijms-24-04614-f005:**
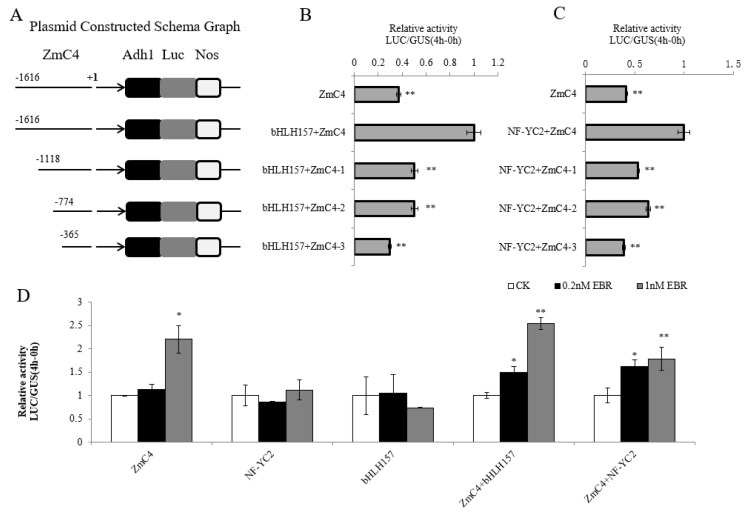
Preliminary exploration of interaction regions between ZmbHLH157/ZmNF-YC2 and ZmC4 promoter. (**A**) Schematic diagram of the construction of a promoter fragment deletion vector. (**B**) Relative activity of LUC after co-transformation of ZmbHLH157 with promoters of different lengths. (**C**) Relative activity of LUC after co-transformation of ZmNF-YC2 with promoters of different lengths. (**D**) Relative activity of LUC after EBL processing transcription factor transient overexpression of protoplasts. “*” represented as *p* < 0.05, “**” represented as *p* < 0.01.

**Table 1 ijms-24-04614-t001:** Prediction of cis-acting elements of ZmC4 NADP-ME promoter.

cis-Element	Position (−)	Sequence	Predicted Function
E-box	435	GTGCAC	bHLH transcription factor potential binding site
	649	CACGTG	
	669	AACGTG	
	1152	TACGTG	
CCAAT-box	1495	CCAAT	CCAAT transcription factor potential binding site
CAAT-box	350	CAAT	
	844	CAAT	
	603	CAAAT	
	1572		
BRRE	245	CGTGCG	brassinosteroid responsiveness
	239		

## Data Availability

Data are contained within the article or Appendix A.

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
