# Peer review of "Epi-Brassinolide Regulates ZmC4 NADP-ME Expression through the Transcription Factors ZmbHLH157 and ZmNF-YC2"

_ijms, 2023, doi:10.3390/ijms24054614_

Round 1
Reviewer 1 Report (Previous Reviewer 3)
This revised manuscript just revised following reviewers' detailed/specific instructions. The big problem still exists: The English level presented in this manuscript by Gao et al is not up to the publication level for IJMS yet. The data were not presented and explained well, with grammar errors here and there, which may have severely damaged the presentation of the Abstract, Introduction, Materials and Methods, Results, Discussion, and Conclusion.
A simple example: Line 52: "the transcript content of ZmC4 NADP ME in the leaves is 20 times than that of" --> "the transcript content of ZmC4 NADP ME in the leaves is 20 times more than that of".
The title: "Epi-brassinolide regulating ZmC4 NADP-ME expression through the transcription factors of ZmbHLH157 and ZmNF-YC2"--> "Epi-brassinolide regulating ZmC4 NADP-ME expression through the transcription factors ZmbHLH157 and ZmNF-YC2".
Author Response
We are glad to receive your comments. We are working hard to improve our manuscript. First, we have corrected other issues raised, and then we will further improve the English level of the paper.
Reviewer 2 Report (Previous Reviewer 1)
Since the MS has revised according to my comments, I suggest that it can be accepted now.
Author Response
We're glad to receive your comments and get your approval.
Reviewer 3 Report (New Reviewer)
The manuscript investigated Epi-brassinolide regulating ZmC4 NADP-ME expression through the transcription factors of ZmbHLH157 and ZmNF-YC2. The authors get the candidate from RNA-seq data and then get the ZmbHLH157 and ZmNF-YC2 by Pearson correlation coefficients and then validate the TF characteristic and confirm the promotor region for binding and further provide insight into BR hormone in maize. The manuscript must be improved. Please see the comments below and try to improve the manuscript:
General comments:
1, The manuscript is short and needs to fully describe the background in the introduction, and the abstract does not clear enough, as well as the discussion.
2, All the figures seem stretched, and the front was messed up; it is also hard to read the figures with the current legend. For example, in figure 2, some of the EBL is uM, and some of them are nM. Also, the label is weird and is not scientific style. It is my first time seeing a figure with EBL VS CK; it might be raw data from the company. Furthermore, all the figures and tables are very poor; please try to make them clear and better present a scientific style.
3, The PCA of the ZmC4 NADP-ME and bHLH157 or NF-YC2 were 0.66, and 0.75 is not strong enough to select these two TFs. Also, the bHLH and NF-YC2 are already known TFs, so it is not necessary to do functional verification, like subcellular localization and self-activation activity, and the structure here does not show anything, the bHLH can form homodimer is known already, and also where the disulfide bonds in your Figure 4C is? Also, why check for the interaction?
4, The reference is messed up. Some of them are in the wrong place and some of them are not enough to support (see details in minor revision); please check them one by one.
5, There is no plant work to support the mechanisms
Minor revision
Line 17, should be brassinosteroid(BR), not BL
Line 20, which may participate in regulating photosynthesis in maize
Line 24, Further experiments showed that
Line 24-25, This sentence is not clear: ZmNF-YC2 and ZmbHLH157 transcription factor binding sites on the ZmC4 NADP-ME promoter were located between –1616 bp and –1118 bp of the target promoter and promoted LUC gene expression
Line 29: BR hormones
Line 32: crop
Line 33: one of the ways?
Line 52: transcript content you mean transcription level?
Line 75: To determine the genes regulated by EBL in maize seedlings, DEGs were analyzed following EBL treatment [36]. Seems the ref 36 is not what you mentioned here
Line 91: Previous experiments have shown that exogenous spraying of EBL promotes the transcription of dark reaction enzymes. Missing ref here
Line 101: Figure2 the front is not consistent
Line 111: EBL is not a signal
Line 115: Pearson correlation coefficients (PCCs, R) what is R stand for?
Line 144: the tertiary structure of bHLH157 was a homodimer structure with disulfide bonds (Figure 4C, left); how can you get this conclusion, and what is the meaning of the predicted structure here
Line154-155: please indicate as much as you can in the figure legend, what is P and R mean? Where is the DNA ladders
Line 197-20: About BES1 and BZR1 as a transcription factor in BRs signal transduction, here is missing ref. There are some studies for maize BES1 and BZR1 binding E-Box (https://link.springer.com/article/10.1007/s10725-018-0424-2; DOI: 10.3390/ijms21030996 ; DOI: 10.1093/jxb/eraa544 )
Line 261: fully describe how you take the leave, did you cut it and set it on MS with EBL and then for RNA-Seq? method part need improvement
Author Response
Point 1: figure 2, some of the EBL is uM, and some of them are nM.
Response 1: These units are all change to nM.
Point 2: The label is weird and is not scientific style. It is my first time seeing a figure with EBL VS CK; it might be raw data from the company.
Response 2: This label have been deleted.
Point 3: The PCA of the ZmC4 NADP-ME and bHLH157 or NF-YC2 were 0.66, and 0.75 is not strong enough to select these two TFs.
Response 3: Several EBL-induced transcription factors were selected, but it was verified by experiments that the expression of these two transcription factors changed the most after EBL treatment, and the remaining transcription factors were excluded after the yeast experiment was verified that the binding ability of the target promoter was not found.
Point 4: There is no plant work to support the mechanisms.
Response 4: Transgenic maize lines are being carried out, but overexpressed maize lines are not easy to transform.
Point 5: Line 24-25, This sentence is not clear: ZmNF-YC2 and ZmbHLH157 transcription factor binding sites on the ZmC4 NADP-ME promoter were located between –1616 bp and –1118 bp of the target promoter and promoted LUC gene expression.
Response 5:This sentence have changed to ZmNF-YC2 and ZmbHLH157 transcription factor binding sites on the -1616 bp and -1118 bp ZmC4 NADP-ME promoter.
Point 6: Previous experiments have shown that exogenous spraying of EBL promotes the transcription of dark reaction enzymes. Missing ref here
Response 6: References related errors have been corrected in this manuscript.
Point 7:Line 115: Pearson correlation coefficients (PCCs, R) what is R stand for?
Response 7:The correlation coefficient (r) indicate the relationship between the variables, while r2 is the Coefficient of Determination and represents the the percentage that the variation of the independent variables contribute in the variation of the dependent Variable.
Point 8: Line154-155: please indicate as much as you can in the figure legend, what is P and R mean? Where is the DNA ladders.
Response 8: The text description has been added. P is the abbreviation for pollen, R means roots. This primer is used for fluorescence quantitative PCR, which is only 200 bp. Because the same pair of primers are used in same gene, the band size is the same, so DNAladder is deleted.
Point 9: Line 261: fully describe how you take the leave, did you cut it and set it on MS with EBL and then for RNA-Seq? method part need improvement.
Response 9: The two leaves of a one-leaf seedling are lightly divided by a sharp blade into segments of about 2cm. This sentence has added in methods.
Round 2
Reviewer 1 Report (Previous Reviewer 3)
I see a revised version. I would like to see a version with tracked changes so I know what has been revised.
Thanks,
Author Response
Dear Reviewer,
We have received your request.
As the original text has been revised, we will send you the revised annotated manuscript with auxiliary explanation,and you can find it in the attachment.
Then,We will then choose MDPI's Language Editing Services to improve the manuscript's English proficiency after confirming that the manuscript is acceptable.
Kind regards.

Reviewer 3 Report (New Reviewer)
Dear authors,
let me remind you again what I point out for the improvement of this paper. please try to figure them out one by one and take it seriously to make it better.
The manuscript investigated Epi-brassinolide regulating ZmC4 NADP-ME expression through the transcription factors of ZmbHLH157 and ZmNF-YC2. The authors get the candidate from RNA-seq data and then get the ZmbHLH157 and ZmNF-YC2 by Pearson correlation coefficients and then validate the TF characteristic and confirm the promotor region for bindingand further provide insight into BR hormone in maize. The manuscript must be improved. Pleasesee the comments below and try to improve the manuscript:
General comments:
1, the manuscript is short and needs to fully describe the background in the introduction, and the abstract does not clear enough, as well as the discussion.
i did not see the reply
2, all the figures seem stretched, and the front was messed up; it is also hard to read thefigures with the current legend. For example, in figure 2, some of the EBL is uM, and some ofthem are nM. Also, the label is weird and is not scientific style. It is my first time seeing a figure with EBL VS CK; it might be raw data from the company. Furthermore, all the figures and tables are very poor; please try to make them clear and better present a scientific style.
it seems you modify the figures but not all, see figure1 B upper left corner
3, The PCA of the ZmC4 NADP-ME and bHLH157 or NF-YC2 were 0.66, and 0.75 is not strong enough to select these two TFs. Also, the bHLH and NF-YC2 are already known TFs, so it is not necessary to do functional verification, like subcellular localization and self-activation activity, and the structure here does not show anything, the bHLH can form homodimer is known already, and also where the disulfide bonds in your Figure 4C is? Also, why check for the interaction?
where is your reply
4, The reference is messed up. Some of them are in the wrong place and some of them are not enough to support (see details in minor revision); please check them one by one.
5, there is no plant work to support the mechanisms
Transgenic maize lines are being carried out, but overexpressed maize lines are not easy to transform. this is not answer the question, you can say science is difficult,
Minor revision
Line 17, should be brassinosteroid(BR), not BL
Line 20, which may participate in regulating photosynthesis in maize
Line 24, Further experiments showed that
Line 24-25, This sentence is not clear: ZmNF-YC2 and ZmbHLH157 transcription factor binding sites on the ZmC4 NADP-ME promoter were located between –1616 bp and –1118 bp of the target promoter and promoted LUC gene expression
Line 29: BR hormones
Line 32: crop
Line 33: one of the ways?
Line 52: transcript content you mean transcription level?
Line 75: To determine the genes regulated by EBL in maize seedlings, DEGs were analyzed following EBL treatment [36]. Seems the ref 36 is not what you mentioned here
Line 91: Previous experiments have shown that exogenous spraying of EBL promotes the transcription of dark reaction enzymes. Missing ref here
Line 101: Figure2 the front is not consistent
Line 111: EBL is not signal
Line 115: Pearson correlation coefficients (PCCs, R) what is R stand for?
Line 144: the tertiary structure of bHLH157 was a homodimer structure with disulfide bonds (Figure 4C, left); how can you get this conclusion, and what is the meaning of the predicted structure here
Line154-155: please indicate as much as you can in the figure legend, what is P and Rmean? Where is the DNA ladders
Line 197-20: About BES1 and BZR1 as a transcription factor in BRs signal transduction,here is missing ref. There are some studies for maize BES1 and BZR1 binding E-Box(https://link.springer.com/article/10.1007/s10725-018-0424-2; DOI: 10.3390/ijms21030996 ; DOI: 10.1093/jxb/eraa544 )
Line 261: fully describe how you take the leave,did you cut it and set on MS with EBL andthen for RNA-Seq?
Author Response
Question 1: the manuscript is short and needs to fully describe the background in the introduction, and the abstract does not clear enough, as well as the discussion.
Response 1: By carefully combing through the logic in the abstract, we added the description of the transcriptome result part, but also gave a clearer description of the result part, specifically described as follows:In this study, transcriptome sequencing of maize seedlings treated with epi-brassinolide (EBL) showed that differentially expressed genes (DEGs) were significantly enriched in photosynthetic antenna proteins, porphyrin and chlorophyll metabolism and photosynthesis pathways. The DEGs of C4-NADP-ME and pyruvate phosphate dikinase in C4 pathway were significantly enriched in EBL treatment. Co-expression analysis showed that the transcription level of ZmNF-YC2 and ZmbHLH157 transcription factors were increased under EBL treatment and were moderately positively correlated with ZmC4-NADP-ME. Transient overexpression of protoplasts revealed that ZmNF-YC2 and ZmbHLH157 activate C4-NADP-ME promoters.
In background, We added a brief description of the mechanism of EBL regulation of photosynthesis. The specific content is as follows: In fact, EBL regulates photosynthesis through different pathways. EBL affects stomatal development by regulating the bHLH family transcription factor SPEECHLESS, and further have an impact on plant photosynthesis. BES1(BRINSENSITIVE 1-EMS-SUPPRESSOR1) mediated the regulation of EBL on chloroplast development [19, 20]. However, the regulation of transcription factors mediated EBL on carbon fixation is not clear.
In discussion, the discussion of transcriptome results is added as follows: BR has previously been found to indirectly affect plant photosynthesis by controlling stomatal size and chloroplast development [11, 27]. We found that the DEGs of EBL treatment were enriched in the photosynthetic antenna protein and porphyrin and chlorophyll metabolism pathway of maize leaves. BR affects chlorophyll content in Brassica juncea [11]. And we have previously found that chlorophyll content in maize leaves is affected by EBL treatment [23]. We speculated that EBL might regulate chlorophyll content in leaves by regulating gene expression in metabolic pathways, and indirectly affect plant photosynthesis.
If a detailed description is not yet available, we will try to modify it to make it more complete.
Question 1: The PCA of the ZmC4 NADP-ME and bHLH157 or NF-YC2 were 0.66, and 0.75 is not strong enough to select these two TFs. Also, the bHLH and NF-YC2 are already known TFs, so it is not necessary to do functional verification, like subcellular localization and self-activation activity, and the structure here does not show anything, the bHLH can form homodimer is known already, and also where the disulfide bonds in your Figure 4C is? Also, why check for the interaction?
Response 1: Indeed, the correlation was not strong, but transient expression results of protoplasts showed that some transcription factors with high correlation were not as strong as the two transcription factors in activating target promoters, and the two were also more highly induced by EBL.
Also, the bHLH and NF-YC2 are already known TFs, but this work was done three years ago before the papers on these transcription factors were published, so the function of transcription factors was verified.
Since both transcription factors were found to activate promoters after transient expression of protoplasts, we speculated that there might be interaction, so the interaction was verified.
there is no plant work to support the mechanisms
In fact, studies have shown that overexpression of ZmC4 NADP-ME in tobacco and sorghum can indeed improve plant photosynthesis and overall growth state, but the specific mechanism is still unclear. Therefore, we hope to find molecular evidence of BL regulation of this gene through transcriptome and related experiments.

Round 3
Reviewer 1 Report (Previous Reviewer 3)
I would like to see a version with tracked changes so that I know what has been revised.
"As the original text has been revised, we will send you the revised annotated manuscript with auxiliary explanation,and you can find it in the attachment.
Then, We will then choose MDPI's Language Editing Services to improve the manuscript's English proficiency after confirming that the manuscript is acceptable."
I would like to see the improvement of the manuscript's English proficiency before it is accepted.
Author Response
"We will send you the revised annotated manuscript with auxiliary explanation,and you can find it in the attachment.
We will improvement of the manuscript's English proficiency before it is accepted."
Reviewer 3 Report (New Reviewer)
please try to not answer same question twice
Author Response
"We will send the revised annotated manuscript with accompanying notes, which you can find in the attachment.
We will improve the English language proficiency of the manuscript before it is accepted.”

Round 4
Reviewer 1 Report (Previous Reviewer 3)
Examples are listed of errors/mistakes that I still identified, but many more are not listed here. The authors should read the whole manuscript carefully and revise. Language is still a big problem.
1, title: "Epi-brassinolide regulating ZmC4 NADP-ME expression through the transcription factors ZmbHLH157 and ZmNF-YC"---> Epi-brassinolide regulates ZmC4 NADP-ME expression through the transcription factors ZmbHLH157 and ZmNF-YC.
2, Please read examples from other IJMS papers and revise "Table 1. KEGG pathways are mainly significantly enriched in EBL vs CK".
3, Please revise all figure legends to make them clear and understandable.
For example,
1) revise "Figure 1. DEGs with EBL treatment in RNA--seq."
2) No need to show Figure 1A.
3) Figure 2, please have a overall title for Figure 2.
4) EBL (uM) should not be under Figure 2C, but Figure 2B.
5) What is 0.2, and 0.5 in figure 2C figure annotation?
4, "2.1. Transcriptome data analysis of EBL processed maize leaves"-->
2.1. Transcriptome data analysis of EBL-treated maize leaves
5, "DEGs were enriched using the KEGG pathway analysis."-->The KEGG pathway analysis was performed in DEGs to identify the enriched pathways.
6. Figure 2, Line 120, "at 5th day and 10th day"---> on the 5th and 10th days.
7. Line 124, "bHLH157 and NF-YC2 induced by EBL" ---> bHLH157 and NF-YC2 are induced by EBL.
8. Figure 3E, the picture ratio looks weird than Figure 3C and 3D. Please make sure the picture is not changed.
9. Figure 4D should be labelled as "C". Revise Figure 4 legend.
10. "2.6. EBL further enhances ZmC4 NADP-ME promoter activity by the improved expression of bHLH157 and NF--YC2"---> EBL further enhances ZmC4 NADP-ME promoter activity by increasing bHLH157 and NF--YC2 expression.
Author Response
Response to Reviewer 1 Comments
Question 1, title: "Epi-brassinolide regulating ZmC4 NADP-ME expression through the transcription factors ZmbHLH157 and ZmNF-YC"---> Epi-brassinolide regulates ZmC4 NADP-ME expression through the transcription factors ZmbHLH157 and ZmNF-YC.
Response 1: Regulating has been modified to regulates.
Question 2, Please read examples from other IJMS papers and revise "Table 1. KEGG pathways are mainly significantly enriched in EBL vs CK".
Response 2: The diagram of KEGG pathway is available. In order to avoid repeated presentation of data, I have deleted this table.
Question 3, Please revise all figure legends to make them clear and understandable.
For example,
1) revise "Figure 1. DEGs with EBL treatment in RNA--seq."
2) No need to show Figure 1A.
3) Figure 2, please have a overall title for Figure 2.
4) EBL (uM) should not be under Figure 2C, but Figure 2B.
5) What is 0.2, and 0.5 in figure 2C figure annotation?
Response 3:
1) “DEGs with EBL treatment in RNA--seq” has been revised to “DEGs number and KEGG pathway enrichment with EBL treatment in RNA-seq”.
2) Figure 1A has been deleted.
3)The overall title “Expression levels of photosynthetic related genes under EBL treatment.” has been added.
4) EBL (uM) has been deleted and the units in figures has all changed from μM to nM.
5) This is EBL concentration and it has changed to 200, and 500 nM.
Question 4, "2.1. Transcriptome data analysis of EBL processed maize leaves"-->
2.1. Transcriptome data analysis of EBL-treated maize leaves
Response 4: “EBL processed maize leaves” has changed to “EBL-treated maize leaves”.
Question 5, "DEGs were enriched using the KEGG pathway analysis."-->The KEGG pathway analysis was performed in DEGs to identify the enriched pathways.
Response 5: This sentence has been corrected.
Question 6. Figure 2, Line 120, "at 5th day and 10th day"---> on the 5th and 10th days.
Response 6: "at 5th day and 10th day" has changed to “on the 5th and 10th days”.
Question 7. Line 124, "bHLH157 and NF-YC2 induced by EBL" ---> bHLH157 and NF-YC2 are induced by EBL.
Response 7: “induced” has been changed to “are induced”.
Question 8. Figure 3E, the picture ratio looks weird than Figure 3C and 3D. Please make sure the picture is not changed.
Response 8: The pictured ratio has been rectified.
Question 9. Figure 4D should be labelled as "C". Revise Figure 4 legend.
Response 9: "C" has been revised to "D".
Question 10. "2.6. EBL further enhances ZmC4 NADP-ME promoter activity by the improved expression of bHLH157 and NF--YC2"---> EBL further enhances ZmC4 NADP-ME promoter activity by increasing bHLH157 and NF--YC2 expression.
Response 10: We are grateful for the suggestion. As suggested by the reviewer, the sentences and word has been revised.

Round 5
Reviewer 1 Report (Previous Reviewer 3)
A shock to see the Author's Notes are not in English.
I ONLY checked the points I listed, and found that some of my points were revised correctly, some were not correctly revised, some were not taken care.
Please recheck and polish.
I suggested last time: "3, Please revise all figure legends to make them clear and understandable."
What is the title for Y-axis for Figure 2B? It is missing "EBL concentration".
What is 200, and 500 in figure 2C figure annotation? "Label ck, EBL 200 nM, EBL 500 nM"?
(B) The transcription level of ZmC4-NADP-ME with EBL treatment in field (maize in 8-leaves stage) on the 5th day and 10th days;
(C) Transcription level of the preliminary screened transcription factors treated with EBL (nM).
Figure 4. bHLH157 and NF-YC2 transcription factor functional verification. (A) Subcellular localization. (B) Yeast self-activation, pGBKT7 was negative control and pGBKT7-GAL4 was positive control. (C) Semi-quantitative, (roots, R), stems (St), leaves (leaves, L), filaments (F), anthers (A), pollen (pollen, P), post-pollination seeds (S), embryos (Em), and endosperm (En).
Author Response
Response to Reviewer 1 Comments
Question 1, What is the title for Y-axis for Figure 2B? It is missing "EBL concentration".
Response 1, the title of Y-axis is added.
Question 2, What is 200, and 500 in figure 2C figure annotation? "Label ck, EBL 200 nM, EBL 500 nM"?
Response 21 The text in Figure 2 B has been changed to:The transcription level of ZmC4-NADP-ME with different concentrations (50 nM, 100 nM, 150 nM, 500 nM) of EBL treatment in field (maize in 8-leaves stage) on the 5th day and 10th day; the expression of ZmC4 NADP-ME in the leaves (seedlings in 2 leaves and 1 core stage) with 0 (CK), 200 nM, 500 nM EBL treatment . The label has been added.
Question 3, What is the title for Y-axis for Figure 2B? It is missing "EBL concentration".
Response 3, The Y axis is the relative expression, and the concentration of EBL is on the abscissa of the bar graph.
Question 4, The transcription level of ZmC4-NADP-ME with EBL treatment in field (maize in 8-leaves stage) on the 5th day and 10th days;
Response 4, Days has changed to day.
Question 5, Transcription level of the preliminary screened transcription factors treated with EBL (nM).
Response 5, This units has added in Figure 2C
Question 6, Figure 4. bHLH157 and NF-YC2 transcription factor functional verification. (A) Subcellular localization. (B) Yeast self-activation, pGBKT7 was negative control and pGBKT7-GAL4 was positive control. (C) Semi-quantitative, (roots, R), stems (St), leaves (leaves, L), filaments (F), anthers (A), pollen (pollen, P), post-pollination seeds (S), embryos (Em), and endosperm (En).
Response 6, Duplicate comments have been removed.
Question 7, 2.6. EBL further enhances ZmC4 NADP-ME promoter activity by theincreasing bHLH157 and NF--YC2 expression
Response 7, “the” have been removed.

This manuscript is a resubmission of an earlier submission. The following is a list of the peer review reports and author responses from that submission.
Round 1
Reviewer 1 Report
The present study focuses on the effect of exogenous hormone brassinolide (BL) on expression level of genes related to photosynthesis, aiming to improve the yield of maize. Indeed, it is interesting for readers to further learn the mechanism of co-operation of transcription factors of ZmbHLH157 and ZmNF-YC2. While there are some points need to be concerned.
1. In the title, the abbreviation of "BL" should be revised to "epi-brassinolide";
2. In the Abstract section, the full name of "NADP-ME" and "EBL" should be provided at the first time.
3. The English language should be carefully checked throughout the text. For example, the description of "the transcript content of ZmC4 NADP-ME in the leaves is 20 times that of cytoplas- 53 mic NADP-ME" should be revised to "the transcript content of ZmC4 NADP-ME in the leaves is 20 times than that of cytoplas- 53 mic NADP-ME" (line 53), "the expression of the gene increased by approximately seven times" should be "the expression of the gene increased by approximately 7 times" (line 97), and "the results showed that DEGs in 0.5 μM EBL treatment on 76 1/2MS solid medium for 6 h had 2041 DEGs" should be " The results showed that 2041 DEGs were observed in 0.5 μM EBL treatment,", amongst others.
4. The size of letters and characters in the Figures should be kept the same, such as in Fig. 1.
5. The Fig. 2C should be separated from the Fig. 2, and shown in an individual picture.
6. In the Fig. 2 and Fig. 3, the mark "*" and “**” should be noted or described in the figure title. In addition, the "a, b, and c" should be changed into "*" and “**”.
7. The Conclusion should be concised, some specific results should be deleted.
8. The datasets should be publicly available at NCBI, and the Sequence Read Archive (SRA) accession must be provided before publication.
Reviewer 2 Report
The research reported the regulation relationship between bHLH157/NF-YC2 and NADP-ME, but the experiment is not rigorous, the function of bHLH157/NF-YC2 and NADP-ME in Brassinolide synthesis and photosynthesis need more experimental evidence supports. Moreover, the writing and grammar is poor and full of errors, even the title is not right. It should not be published.
Reviewer 3 Report
This manuscript by Gao et al did RNA-Seq of EBL treatment to maize, together with fluorescence quantification, transcription activation, subcellular localization in protoplast, yeast one-hybrid, and dual luciferase system, to identify the regulation of EBL on ZmC4-NADP-ME.
The English level presented in this manuscript by Gao et al is not up to the publication level yet. The data were not presented and explained well, with grammar errors here and there, which may have severely damaged the presentation of the Abstract, Introduction, Materials and Methods, Results, Discussion, and Conclusion. It is suggested that the manuscript should have careful proofreading and language editing (for example, collaborating scientists or journal manuscript editing service) and can be re-submitted once the English level is improved to the publication level.
Please read the IJMS papers and follow the formatting.
Examples are listed here, but not limited to:
1) Title: “EBL regulating ZmC4 NADP-ME expression though the transcription factors of ZmbHLH157 and ZmNF-YC2”
You mean here EBL regulating ZmC4 NADP-ME expression through the transcription factors of ZmbHLH157 and ZmNF-YC2?
22) EBL or EBR? Please be consistent. Both show in the manuscript.
33) Line 22: “The results showed that the two transcription factors were increased under EBL treatment”
You mean: The results showed that the transcript levels of these two transcription factors were increased under EBL treatment?
44) Lines 75-77: “To determine the genes regulated by EBL in maize seedlings, DEGs were analyzed following EBL treatment. The results showed that DEGs in 0.5 µM EBL treatment on 1/2MS solid medium for 6 h had 2041 DEGs…..” needs revision.
55) Figure legends need revision, figures need necessary labels to help the readers to understand the figures.
66) Figure 1, “(B) Yolcanoplot”à Volcanoplot?
77) Figure 2, “at 5th day and 10th day”à on (a day). What are the definition of “*”, “**”?
88) Figure 3E, what are the six columns?
99) Lines 91-92: “Previous experiments have shown that exogenous spraying of EBL promotes the transcription of dark reaction enzymes.” à Cite reference(s).
110) Line 150: “(C) Protein tertiary structure”àpredicted structure? By what software?
111) Line 182-183: “2.6. EBL promotes the expression of bHLH157 and NF-YC2 to active ZmC4 NADP-ME promoter”à What does active mean here?
Reviewer 4 Report
In this manuscript, Gao et al evaluated the Brassinolide’s (in the form of EBL) effect on ZmC4 NADP-ME, a key enzyme of the C4 cycle. They identified the ZmNF-YC2 and ZmbHLH157 as two strong candidate genes from the RNA-Seq analysis. The authors did the Y1H and Luciferase assays to confirm that ZmNF-YC2 and ZmbHLH157 transcriptionally regulate ZmC4 NADP-ME.
Although the authors provide new knowledge about the transcriptional regulation of ZmC4 NADP-ME, I have some major and minor concerns about the manuscript.
Major:
1. The manuscript needs a lot of details. To name a few, a short description of the experiment in Result section 2.1; the details about the protoplast assay and how it was conducted; the details about the RNA-Seq analysis (how DE genes were identified), etc.
2. In Result 2.1, Please explain why to use 0.5uM concentration EBL treatment. Why use the solid medium instead of hydroponic? What stage is the maize used?
3. All the tables and figure legends miss a lot of key information, which is hard to interpret.
4. The experiments in the Result section need some details, otherwise, it’s really hard to follow the logic.
5. How ZmbHLH157 and ZmNF-YC2 stand out as two candidates are not clear. Are these two TFs with the highest PCC with ZmC4 NADP-ME among DE genes? Please provide a table for log2FC and PCC for DE genes.
6. The yeast experiment in Result 2.3 needs details. So does Result 2.4.
7. Figure 4D is not very convincing. Please use a better gel picture. Preferable to be on the same gel. If the experiment is hard to re-do, I recommend authors use the MaizeGDB's expression browser.
8. Figure 5s are confusing. Please explain with more details.
9. On line 187, “increased by 1.55 and 0.79 times”. Not sure what 1.55 and 0.79 refer to. Does “0.79 times” still an increase?
10. The RNA-Seq data should be uploaded to NCBI-SRA for public usage.
Minor:
1. The “though” in the title should be “through”.
2. GeneIDs for ZmC4 NADP-ME, ZmNF-YC2 and ZmbHLH157 should be provided in the abstract.
3. On line 52, “the three NADP-MEs are similar”. They are “similar” in what aspects? Is the sequence similar? If so, please provide multi-sequence alignment as a supp.
4. Many misspells and grammar errors in the manuscript.
5. Figure 1B, please use another set of colors other than “Red-Green” for color-blind readers.
6. Figure 1C, what is the "Rich factor"? The color scale is confusing. Please use a sequential color for qvalue <0.05 (https://colorbrewer2.org/#type=sequential&scheme=BuGn&n=3). Qvalues above 0.05, should be grey or black.
7. Figure 2A, please add gene ID. The text is truncated.
8. Figure 2B, what is DAP? Day after pollination/planting? Please justify why only 150nM-5DAP and 100nM-10DAP were upregulated.
9. On line 154, need a reference for Plant Care.
10. Please present Table 2 as a diagram for better understanding.